# Picture semantic similarity search based on bipartite network of picture-tag type

**Mingxi Zhang**[1,2]*, **Liuqian Yang**[1], **Yipeng Dong**[1], **Jinhua Wang**[2], **Qinghan Zhang**[1]

**1** College of Communication and Art Design, University of Shanghai for Science and Technology, Shanghai, China, **2** School of Computer Science, Fudan University, Shanghai, China

* mingxizhang10@fudan.edu.cn

## Abstract

Searching similar pictures for a given picture is an important task in numerous applications, including image recommendation system, image classification and image retrieval. Previous studies mainly focused on the similarities of content, which measures similarities based on visual features, such as color and shape, and few of them pay enough attention to semantics. In this paper, we propose a link-based semantic similarity search method, namely PictureSim, for effectively searching similar pictures by building a picture-tag network. The picture-tag network is built by "description" relationships between pictures and tags, in which tags and pictures are treated as nodes, and relationships between pictures and tags are regarded as edges. Then we design a TF-IDF-based model to removes the noisy links, so the traverses of these links can be reduced. We observe that "similar pictures contain similar tags, and similar tags describe similar pictures", which is consistent with the intuition of the SimRank. Consequently, we utilize the SimRank algorithm to compute the similarity scores between pictures. Compared with content-based methods, PictureSim could effectively search similar pictures semantically. Extensive experiments on real datasets to demonstrate the effectiveness and efficiency of the PictureSim.

**Data Availability Statement:** All relevant data are within the manuscript. The ImagNet data is available from the ImagNet website http://www.image-net.org/, which is organized according to the WordNet hierarchy (currently contains only the

## Introduction

Searching similar pictures for a given picture is an important task in numerous applications. Typical examples include medical image classification [1], image forgery detection [2], image recommendation system [3] and image cluster analysis [4], in which picture plays an important role. Traditional picture similarity search methods compute similarities based on visual features, namely Content-based Image Retrieval (CBIR), including color and shape, *e.g.*, GIST [5], SIFT [6] and SURF [7]. [8] aggregates local deep features to produce compact global descriptors. [9] proposes FTS to get fractal-based local search. [10] proposes VWIaC and FIbC to build smaller and larger sizes of codebooks for salient objects within pictures. [11] proposes Iterative Search (IS) to achieve search similar pictures effectively, which extracts knowledge from similar pictures to compensate for the missing information in the feature extraction process. [12] measures similarities between pictures by using distributed environments and LSH in the distributed scheme. [13] employs an improvement of the D-index method to reduce

nouns). ImagNet data is widely used in advancing computer vision and deep learning research. The Nipic data is available from the Nipic website http://www.nipic.com/index.html, which is a sharing platform for picture materials. We crawled the pictures with tag information from the website for building the picture-tag network. As their website states, both ImagNet and Nipic are available for free to researchers for non-commercial use.

**Funding:** This work was supported by National Natural Science Foundation of China under Grant 62002225, and Natural Science Foundation of Shanghai under Grant 21ZR1445400. The funders had no role in study design, data collection and analysis, decision to publish, or preparation of the manuscript.

**Competing interests:** The authors have declared that no competing interests exist.

computational overhead in large, high-dimensional and scalable picture datasets. [14] assesses similarities between pictures based on the high dimension biomimetic information geometry theory and interactive fork value method. [15] uses the fusion of SIFT and BRISK visual words to get similar pictures in terms of content. IMCEC [16] employs a deeper architecture of CNNs to provide different semantic representations of the picture, which makes it possible to extract features with higher qualities. [17] proposes a hybrid PCA–whale optimization-based deep learning model for the classification of picture, including transform picture dataset by one-hot encoding approach, reduce the dimensions of the transformed data by PCA and select the optimal features by WOA. [18] discusses the application of DL in medical image processing, which could realize the tracking, diagnosis and treatment of virus spread. [19] proposes an effective ensemble learning approach to identify and detect objects, which could achieve good accuracy on both with-in as well as cross-corpus datasets. [20] proposes a deep learning-based object detection approach, which utilizes ResNet to achieve fast robust and efficient object detection.

Practically, the measurement of picture similarity should be based on semantic information rather than visual features, which could cause a "semantic gap" between "semantic similarity" by human judgments and "visually similarity" by computer judgments. More precisely, picture semantic similarity is to answer the question "how similar are these two pictures?". For example, if there are two pictures with different colors and backgrounds in "Cell phone" advertisements, which should be considered to be similar semantically, but they might be treated as dissimilar in visual features. On the other hand, different semantic pictures with similar visual features are judged to be similar pictures by the content-based methods. Due to the lack of semantic consideration, content-based metrics mainly focus on finding similar pictures in terms of visual features rather than semantics, which might neglect the expected similar pictures and deviate from the user's intention. Though it is non-trivial to find an alternative model to search similar pictures semantically. Fortunately, we observed that the semantic information of a picture can usually be described by several tags. For example, "Cell phone" pictures can be described by "Cell", "iPhone", "Huawei", "Mobile" and so on. This motivates us to build an alternative model for searching similar pictures based on "description" relationships between pictures and tags.

As mentioned in [19, 22], the returned result of link-based similarity measures could produce a better correlation with human judgments compared with content-based methods. Therefore, it is reasonable to believe that search similar pictures based on links is worth thoroughly exploring. Among link-based metrics, SimRank [21] is one of the most influential ones, due to it relies on the simple intuition that objects are similar if they point to similar objects. Moreover, it can capture structural similarity very well, because it no longer only considers direct in-links among nodes but also indirect in-links. And SimRank is a universal model, it builds a network by objects-to-objects relationships, whose similarity could be defined based on the structural context of objects.

There are also other metrics of this kind. P-Rank [22] is similar to SimRank, but it improves SimRank by incorporating both in-links and out-links between objects and controls the relative importance of in/out-links through a parameter. PathSim [23] calculates the similarity between objects by the numbers of meta-paths on a heterogeneous network. SimCat [24] defines the similarities between objects by incorporating category information and aggregating relationship network structures. SLING [25] uses an index for storing hitting probabilities, which answers single-source SimRank query with a worst-case error in each SimRank score. TSF [26] builds one-way graphs by randomly sampling one in-neighbor for each node in a given graph, and finds similar nodes based on the one-way graph. ProbeSim [27] assesses similarity without precomputing index structure, so it can support real-time computation of top $k$

**Table 1. Summary of related studies on searching similar pictures.**

| References | Dataset | Methods used | Evaluation metrics | Limitations |
|---|---|---|---|---|
| [8] | INRIA Holidays dataset, Oxford buildings dataset, Oxford buildings dataset+100K and University of Kentucky Benchmark dataset | SIFT, fisher vectors and triangulation embedding | Ratio to median, dimensionality and overfitting effect | Model does not consider searching similar pictures in terms of semantic |
| [11] | Oxford Buildings, Object Sketches, a large-scale dataset collected by author | HOG, HOF, GIST and convolutional neural network (CNN) | Accuracy, NDCG@$k$ and Precision@$k$ | Itertive search needs expensive overhead |
| [16] | Malimg dataset | VGG16, ResNet-50 and Support Vector Machine (SVM) | Accuracy, precision, recall, F1-score, true positive rate (TPR) and false positive rate (FPR), | Model takes expensive time overhead |
| [17] | Plant–village dataset repository | PCA and WOA | Accuracy, Loss and time | Results confined to only dataset in tomato plant diseases |
| [26] | SNAP, KNOECT and LWA | Monte Carlo and approximation random model | Precision, NDCG@$k$ and time cost for building index | TSF does not provide a guarantee of the worst accuracy |
| [25] | SNAP and LWA | Monte Carlo and linearization method | Maximum error, average error, precision, preprocessing time and space consumption | Sling can't effectively update in dynatic network |
| [27] | SNAP and LWA | Index-free SimRank and random walks | Precision@$k$, NDCG@$k$, $\tau_k$ and query time | ProbeSim does not limit random walks lengths |
| [35] | SNAP and DBLP | Random walks and SimMaps | Accuracy ratio and loss, P@$k$, Kendall Tau difference and running time | TopSim does not limit random walks lengths |

queries on a dynamic network. SimRank* [28] remedies the problem of "zero-similarity" in SimRank, which enriches semantics without suffering from increased computational overhead. PRSim [29] is based on the main concepts of SLING, which leverages the graph structure of power-law to efficiently answer SimRank queries, and a connection is built between SimRank and personalized PageRank. UniWalk [30] calculates the similarities between objects based on Monte Carlo, which could directly locate the top $k$ similar vertices for any single source via $R$ sampling paths originating from the single source. SimPush [31] speeds up query processing by identifying a small number of nodes, then computes statistics and performs residue push from these nodes. These measures have been applied in numerous applications, such as spam detection [32], web page ranking [33], citation analysis [34].

Table 1 summarizes several picture similarity search methods, including content-based and link-based. Compared with the latest content-based metrics, link-based similarity measures could capture the semantic information of pictures based on a picture-tag network, while content-based methods mainly focus on searching similar pictures in visual features, which might neglect the expected similar pictures and deviate from the user's intention. Moreover, the intuition of link-based methods is that "two pictures are similar if they are related to similar pictures", which could search underlying similar pictures. For example, picture A is similar to picture B, and picture A is similar to picture C, so picture B is similar to picture C.

In this paper, we propose a link-based picture semantic similarity search method, namely PictureSim, for effectively searching similar pictures by building a picture-tag network. We first build a picture-tag network based on "description" relationships between pictures and tags, and then exploit the object-to-object relationships [36, 37] in picture-tag network. The intuition behind PictureSim is that "similar pictures contain similar tags, and similar tags describe similar pictures", which is consistent with the intuition of SimRank. Consequently, we adopt SimRank model [21] to compute the similarity scores, which helps to find underlying similar pictures semantically.

Our main contributions are as follows.

- We build a picture-tag network by "description" relationships between pictures and tags. Initially, tags and pictures are treated as nodes, and relationships between pictures and tags are regarded as edges. Then, we propose a TF-IDF-based method to remove the noisy links by setting a threshold, which could measure whether a tag has good classification performance.

- We propose a link-based picture similarity search algorithm, namely PictureSim, for effectively searching similar pictures semantically, which considers the context structure to search underlying similar pictures in a network. And it could respond to the user's requirement timely.

- We ran a comprehensive set of experiments on Nipic datasets and ImageNet datasets. Our results show that PictureSim achieves semantic similarity search between pictures, which produces a better correlation with human judgments compared with content-based methods.

## Methods

In this section, we show a framework of the top $k$ picture semantic similarity search, which is divided into two stages. The first stage is to build a picture-tag network by "description" relationships between pictures and tags, in which pictures and tags are regarded as nodes, and relationships between the pictures and the tags are regarded as edges. And we remove the noisy links based on TF-IDF model, in which a few informative tags are removed. Then, we use Sim-Rank algorithm to search the top $k$ most similar pictures for a given picture.

Compared with content-based methods, PictureSim can achieve semantically similarity by building a picture-tag network, while content-based methods can only achieve visual similarity. And users usually judge similarities based on semantics rather than visual features.

### Problem definition

For subsequent discussions, we first give the definition of the top $k$ picture semantic similarity search, that is defined as:

**Definition 1.** *Top-k picture semantic similarity search In the picture-tag network, given query q in the network, a positive integer $k < n$, a top-k semantic similar picture is to find k most similar pictures in terms of semantic and ranked with similarity descending.*

### Network building

**Definition of picture-tag network.**   Tags are descriptive keywords to discriminate objects. For example, the web tag is a way to organize Internet content. It helps users classify and describe the content of web retrieval. The purpose of tag generation is to find semantic information of a given object. There is a need for finding semantic information of objects. Thus, many approaches to generate tags are developed, including user annotation and machine generation. For example, Oriol et al. [38] proposed an attention mechanism, which maps each word of the picture-generated description to a certain area of the picture. Therefore, there is semantic information between the tag and the picture, which provides an important guarantee for semantic similarity computation. The review network [39] as an extension of [38], it can learn the annotation and initial states for the decoder steps. In the picture-tag network, tags could fully express the semantic information of pictures, which helps search similar pictures semantically. The picture-tag network is defined as:

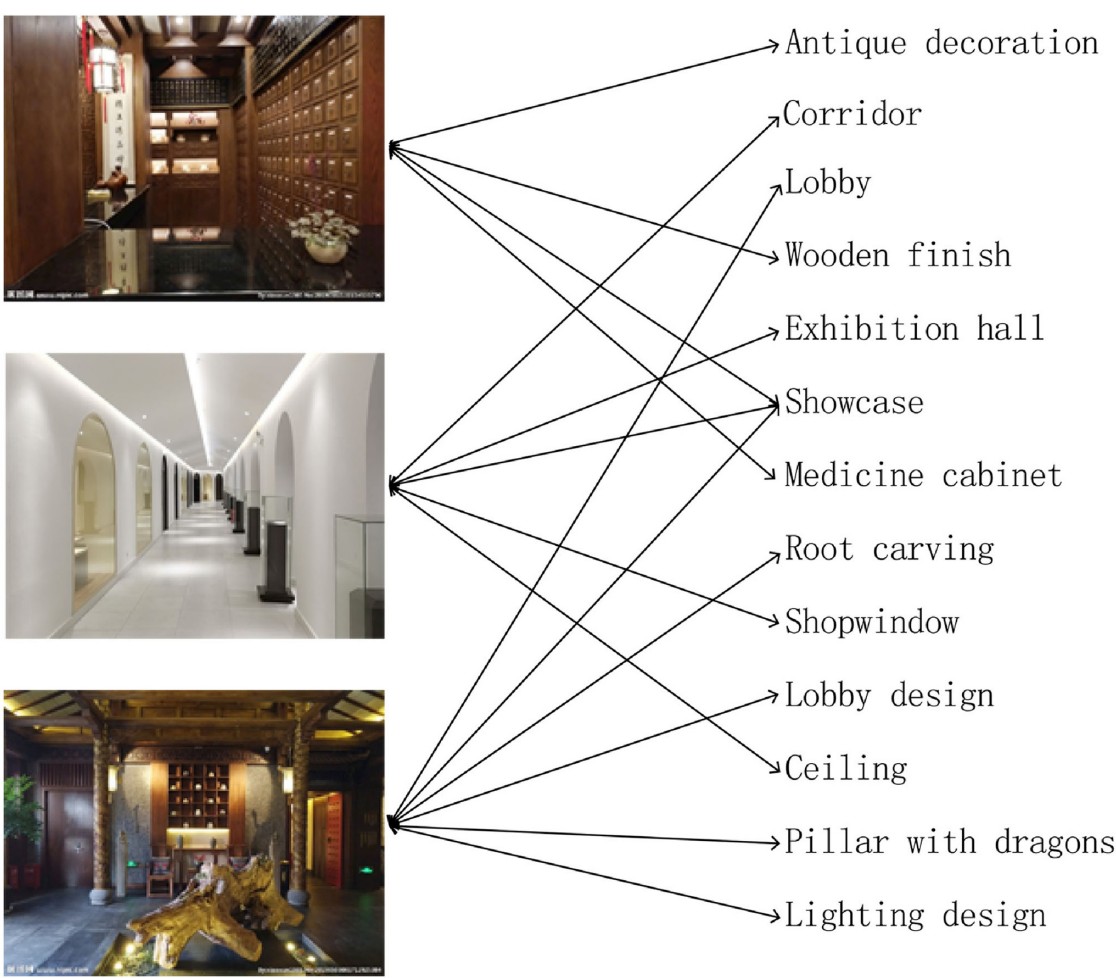

**Fig 1. Example of picture-tag network.**

**Definition 2.** *Picture-Tag Network* *A picture-tag network is defined as a bipartite network G = (V, E), where $V = V_P \cup V_T$, and $V_P$ and $V_T$ represent the sets of pictures and tags respectively; E denotes the set of edges of "description" relationship between pictures and tags, and an edge e $(p_i, t_j) \in E$ denotes a picture $p_i \in V_P$ is described a tag $t_j \in V_T$.*

In a picture dataset, a "description" relationship between a picture and a tag builds an link, and all of the "description" relationships build a picture-tag network. Fig 1 is a toy picture-tag network, pictures and tags are treated as nodes, and "description" relationships between the pictures and the tags are treated as edges. Fig 1 shows that the first picture is described by some tags, including "antique decoration", "wooden finish", "showcase", etc. A link between the first picture and "antique decoration" could represent a "description" relationship, and a "description" relationship and "be described" relationship exist simultaneously. Similarly, a tag can also describe several pictures, such as "showcase" describes three pictures. These tags can fully illustrate semantic information of pictures, which has a better correlation with human judgments in similarity search.

**Removing noisy links.** Noisy links are the tags that cannot effectively discriminate pictures when computing the similarities. It not only affects search results but also incur the expensive time and space overhead, so it is necessary to be removed. Term Frequency-Inverse

Document Frequency (TF-IDF) [40] could be seen as a promising method to find noisy links. It is a statistical method to assess whether a tag is important for a picture. In other words, if a tag describes a picture, and this tag rarely appears in the description of other pictures. It indicates that the tag has a better discrimination ability. Term frequency (TF) indicates how often a tag $t$ appears in a picture $p$, it is defined as:

$$\text{tf}_{t,p} = \frac{n_{t,p}}{|O(p)|} \tag{1}$$

where $n_{t,p}$ is the number of a tag $t$ is associated with a picture $p$, and we set $n_{t,p}$ roughly as 1. There is no duplicate tags $t$ will appear in the description of the picture $p$, because the tag is different from the text; and $|O(p)|$ is the number of out-neighbors of picture $p$. The inverse document frequency (IDF) is a measure of the universal importance of a tag, it is defined as:

$$\text{idf}_{t} = \frac{\log|n_p|}{|I(t)|} \tag{2}$$

where $|n_p|$ is the total number of pictures in the data, $|I(t)|$ is the number of in-neighbors of tag $t$. Based on TF and IDF, the TF-IDF value for tag $t$ and picture $p$ is defined as:

$$\text{tfidf}_{t,p} = \text{tf}_{t,p} * \text{idf}_{t} \tag{3}$$

Intuitively, the tag has good recognition performance if tags have high TF-IDF. And tags with lower TF-IDF should be removed to avoid affecting results. Then, we remove noisy links with low TF-IDF according to a threshold $\delta$, defined as $\delta = (\max - \min) * h + \min$ where $h = (0, 1)$. In a picture-tag network, the links correspond to TF-IDF values lower than $\delta$ are removed before similarity computation.

## Similarity model

Link-based approaches search similar pictures semantically by building a picture-tag network. And SimRank [21] can be regarded as one of the most attractive methods. Because it no longer only considers direct in-links among nodes but also indirect in-links. Then SimRank is a general model that can be applied in any similarity search field, and it is suitable for bipartite networks. There are some other link-based similarity measures, such as PageSim [41], P-Rank [22] and SimRank* [28]. P-Rank enriches SimRank by jointly encoding both in- and out-link relationships into structural similarity computation. And the picture-tag network is bipartite, which makes both SimRank and P-Rank equivalent. PageSim and SimRank* consider the paths of unequal length to search similar pictures. However, PictureSim only considers the paths of equal length.

PictureSim uses SimRank to compute similarity in a picture-tag network. Our key observation is that "similar pictures contain similar tags, and similar tags describe similar pictures". As shown in Fig 1, the similarity score between the first picture and itself is 1, similarly for "showcase". Clear, three pictures are similar: all have the "showcase", and the reason we can conclude that three pictures are similar is that they are described as "showcase". The first picture is described as "wooden finish" while the second picture is described as "shopwindow", and these are similar tags. In this sense that they describe similar pictures.

Let $S(p_1, p_2)$ denotes the similarity between pictures $p_1 \in V_P$ and $p_2 \in V_P$, and let $S(t_1, t_2)$ denotes the similarity between tags $t_1 \in V_T$ and $t_2 \in V_T$. If $p_1 = p_2$, $S(p_1, p_2) = 1$, and similarly

for $S(t_1, t_2)$. For $p_1 \neq p_2$, $S(p_1, p_2)$ is defined as:

$$S(p_1, p_2) = \frac{c}{|O(p_1)||O(p_2)|} \sum_{i=1}^{|O(p_1)|} \sum_{j=1}^{|O(p_2)|} S(O_i(p_1), O_j(p_2)) \qquad (4)$$

and for $t_1 \neq t_2$, $S(t_1, t_2)$ is defined as:

$$S(t_1, t_2) = \frac{c}{|I(t_1)||I(t_2)|} \sum_{i=1}^{|I(t_1)|} \sum_{j=1}^{|I(t_2)|} S(I_i(t_1), I_j(t_2)) \qquad (5)$$

where $c$ is a constant between 0 and 1, which is typically set as 0.8 according to [21]; $|O(p_1)|$ is the number of elements of the set $O(p_1)$, $|I(t_1)|$ is the number of elements of the set $I(t_1)$, $O(p_1)$ is the number of out-neighbors of picture $p_1$ and $I(t_1)$ is the number of in-neighbors of tag $t_1$. $O_i(p_1)$ denotes the $i$-th out-neighbor of picture $p_1$, and $I_j(t_1)$ denotes the $j$-th in-neighbor of tag $t_1$, where $1 \leq i \leq |O(p_1)|$ and $1 \leq j \leq |I(t_1)|$. If $O(p_1) = \emptyset$ or $O(p_2) = \emptyset$, $S(p_1, p_2) = 0$, and similarly for $S(t_1, t_2)$.

The similarity scores are computed iteratively. At the $l$-th iteration, $R_l(p_1, p_2)$ denotes the similarity scores between picture $p_1$ and picture $p_2$, $R_l(t_1, t_2)$ denotes the similarity scores between tag $t_1$ and tag $t_2$. If $p_1 = p_2$, $R_0(p_1, p_2) = 1$ at $l = 0$, otherwise $R_0(p_1, p_2) = 0$, and similarly for $R_0(t_1, t_2)$. When $l = 2, 3, 4 \ldots$, $R_{l+1}(p_1, p_2)$ is defined as $R_{l+1}(p_1, p_2) = 1$ if $p_1 = p_2$, otherwise:

$$R_{l+1}(p_1, p_2) = \frac{c}{|O(p_1)||(p_2)|} \sum_{i=1}^{|O(p_1)|} \sum_{j=1}^{|O(p_2)|} R_l(O_i(p_1), O_j(p_2)) \qquad (6)$$

and similarly, $R_{l+1}(t_1, t_2)$ is defined as: $R_{l+1}(t_1, t_2) = 1$ if $t_1 = t_2$, otherwise:

$$R_{l+1}(t_1, t_2) = \frac{c}{|I(t_1)||(t_2)|} \sum_{i=1}^{|I(t_1)|} \sum_{j=1}^{|I(t_2)|} R_l(I_i(t_1), I_j(t_2)) \qquad (7)$$

## On-line query processing

Based on Eq (6), the similarities between pictures can be computed in the off-line stage. A straightforward method to find the top $k$ similar pictures is that: we first choose $k$ most similar pictures based on the pre-computing similarity scores, then sort and return them. Though this can save time overhead in on-line stage, expensive operations are required in the off-line stage, which involves $O(n^2)$ time cost and $O(ld^2 n^2)$ space cost at the $l$-th iteration, where $n$ is the number of nodes in the network, $d$ is the average degree of the nodes, and we set $l$ from 1 to 7 in terms of time and cost overhead. Therefore, the computation would become inefficient especially when the picture-tag network grows large.

Fortunately, there is a large portion of optimization techniques on SimRank similarity search, *e.g.*, TopSim [35], Par-SR [26] and ProbeSim [27], which searches similar pictures without any preprocessing, a typical example is TopSim. TopSim focuses on computing exact SimRank efficiently. It uses neighborhood to describe the structural context of a node, then merges certain random walk paths by maintaining a similarity map at each step. Therefore, PictureSim optimizes the efficiency of SimRank by TopSim algorithm without any preprocessing, which requires $O(d^{2l})$ time cost in the on-line stage.

## Results

In this section, experimental results are reported in real datasets. Experiments were done on a 2.3 GHz Intel(R) Core i5 CPU with 8 GB main memory. All algorithms were implemented in Java by using Eclipse Java 2018.

### Datasets and evaluation

In the experiments, we extract picture-tag networks from **Nipic** dataset (http://www.nipic.com/index.html) and **ImageNet** dataset (http://www.image-net.org/) to evaluate our approach. Nipic contains 37,221 pictures, 58,623 tags and 610,440 "description" relationships. The parameter $h$ is set to be 0.8 to remove noisy links if not specified explicitly, and we finally obtain 283,079 links. We select the sub dataset ILSVRC-2012 from ImageNet, which contains 50,000 pictures, 1,000 tags and 50,000 "description" relationships.

We implemented four contrast algorithms to evaluate the effectiveness: SimRank algorithm [21] and some content-based algorithms that include Minkowski Distance (MD) [42], Histogram Intersection (HI) [43] and Relative Deviation (RD). We use TopSim [35] to improve the efficiency of SimRank, which only needs to find candidates from the neighborhood locally without traversing the entire network. TopSim is used in a homogeneous network in [35], and we use it in a heterogeneous network. The decay factor $c$ of SimRank is set as 0.8. The MD defines a set of distances, which makes it possible to measure the distance between points. In the HI, each feature set is mapped to a multi-resolution histogram that preserves each feature's distinctness at the finest level. RD judges the similarities by calculating relative deviation.

In the dataset, we randomly pick 20 pictures to test the effectiveness of different algorithms for the top $k$ query with $k = 50$. Effectiveness is evaluated by Mean Average Precision (MAP), which is formally defined as $\mathrm{MAP} = \sum_{q=1}^{Q} \mathrm{AveP}(q)/Q$, where $Q$ is the number of query pictures and AveP(q) is the average MAP scores of the query picture $q$. MAP scores are computed according to the similarity levels which are set as six levels: 0 (dissimilar), 0.2 (potentially similar), 0.4 (marginally similar), 0.6 (moderately similar), 0.8 (highly similar) and 1 (completely similar). The similarity levels are labeled by people, which is a gold standard due to we judge the semantic similarity of pictures based on users' understanding of the pictures.

### Nipic

Table 2 shows the MAP scores of different metrics in Nicpic, and PictureSim sets $l$ as 5. The MAP scores of PictureSim are obviously higher than that of traditional content-based methods with different $k$. For example at $k = 15$, PictureSim achieves average 0.599 MAE, while RD and HI yield average 0.119 MAE. This is because PictureSim computes similarity scores by the structure of context in the picture-tag network, while the traditional content-based approach considers the visual features, which often fails to reflect the semantic information in the user's mind.

Fig 2(a) shows the MAP scores with varying $l$ in Nipic, which clearly illustrates the effect of $l$ in PictureSim. We observe that the MAP scores increase slowly as $l$ increases from 1 to 5,

**Table 2. MAP at different $k$ in Nipic.**

| K | PS | MD | RD | HI |
|---|-----|-----|-----|-----|
| 5 | 0.718 | 0.258 | 0.264 | 0.256 |
| 10 | 0.655 | 0.162 | 0.158 | 0.161 |
| 15 | 0.599 | 0.123 | 0.119 | 0.119 |

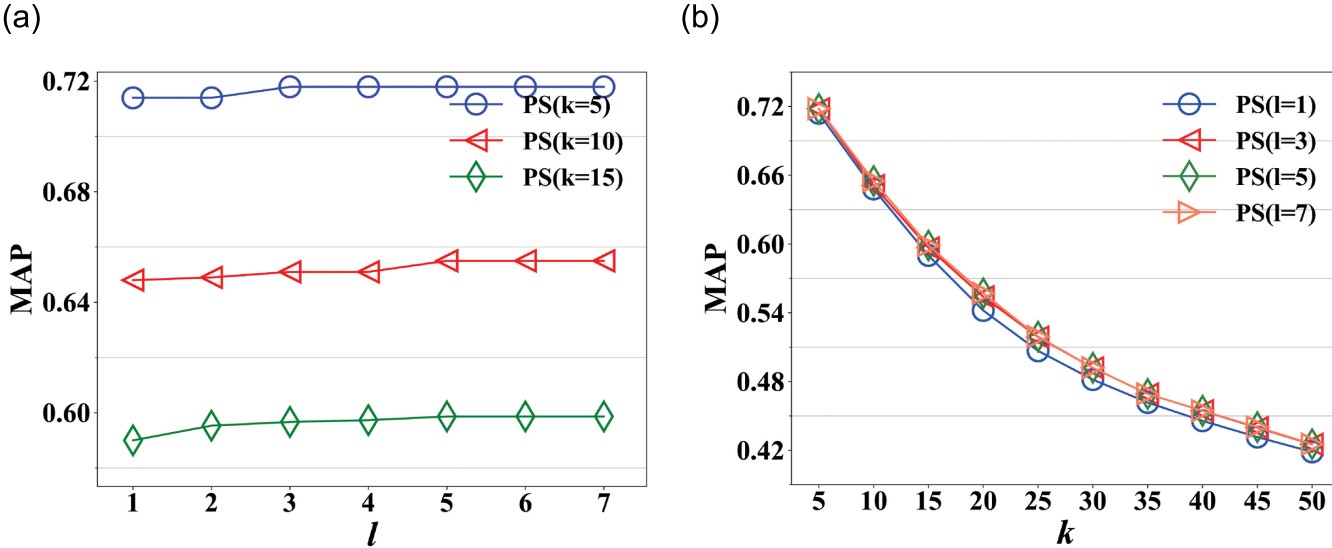

**Fig 2. MAP on varying *l* and *k* respectively in Nipic.** (a) MAP on varying *l*, (b) MAP on varying *k*.

because PictureSim not only considers direct in-links among nodes but also indirect in-links. After $l = 5$, the MAP scores become stable, and PictureSim converges to a stable state. So the returned rankings would become stable empirically after the fifth iteration.

Fig 2(b) shows the MAP scores of PictureSim on varying $k$ in Nipic. The MAP scores gradually decrease as $k$ increases, it could achieve average 0.718 MAP at $k = 5$. This is because the higher similarity scores have a higher rank in the returned list. Generally, users are only interested in the top 10 similar pictures for a given picture, so PictureSim could achieve the user's intention.

Fig 3(a) shows the MAP scores of PictureSim on varying $h$ in Nipic, where $l = 3$. The MAP scores are relatively stable as $h$ increases from 0.1 to 0.4, and increases evidently at $h = 0.5$,

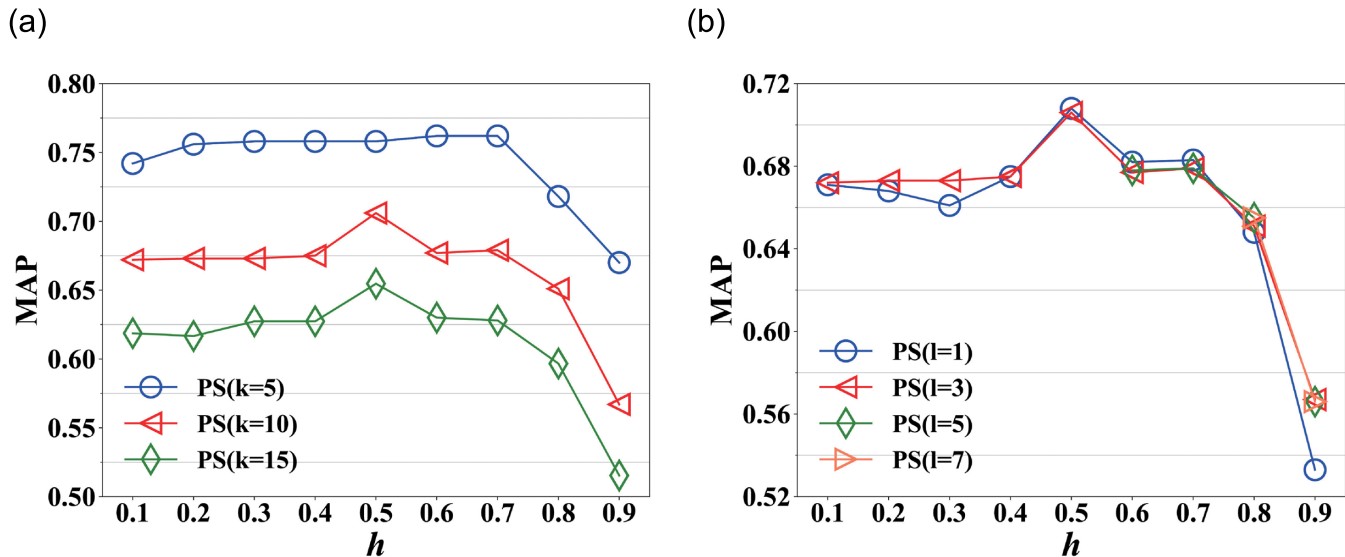

**Fig 3. MAP on varying *h* in Nipic.** (a) MAP on varying *h* at $l = 3$, (b) MAP on varying *h* at $k = 10$.

**Table 3. MAP of top 10 similar pictures from different user perspectives in Nipic.**

| Aspect | PS | MD | RD | HI |
|---|---|---|---|---|
| Semantics | 0.655 | 0.162 | 0.158 | 0.161 |
| Shape | 0.326 | 0.476 | 0.486 | 0.342 |
| Color | 0.294 | 0.405 | 0.397 | 0.361 |

then drops continuously and the curves reach the bottom at $h = 0.9$. This is because the more noisy links are removed if $h$ becomes large. However, some useful links might be also removed as $h$ increases, and consequently the MAP scores decrease. And the curve is an exception at $l = 1$, the MAP scores are relatively stable from $h = 0.1$ to $0.7$, then decreases evidently as $h$ increases, due to it considers direct in-links among nodes, and other curves not only consider direct in-links but also indirect in-links. Similar results can be found in Fig 3(b), which shows the MAP scores of PictureSim on varying $h$ where $k = 10$. And the result can be explained similarly due to the change of MAP is similar with Fig 3(a).

To compare the performance of different algorithms from user's perspectives in Nipic, including semantic, color and shape, we calculate the MAP scores of top 10 similar pictures are shown in Table 3, where $l = 5$ in PictureSim. Obviously, PictureSim has relatively higher MAP scores compared with traditional content-based metrics in terms of semantics, while the comparison methods have relatively higher MAP scores in terms of shape and color. This is because we pay more attention to whether the tag fully expresses the semantics of pictures rather than visual feature, and color and shape often fail to fully express the semantic information of the picture.

Fig 4(a) shows the running time on varying $l$ in Nipic, in which, the running time increases slowly before $l = 5$ and increases rapidly after $l = 5$. This is because PictureSim also considers indirect in-links when searching similar pictures, it needs to traverse more paths as $l$ increases. Fortunately, PictureSim could converge rapidly at $l = 5$ as shown in Fig 2(a), which shows a good performance of the proposed approach.

(a)

(b)

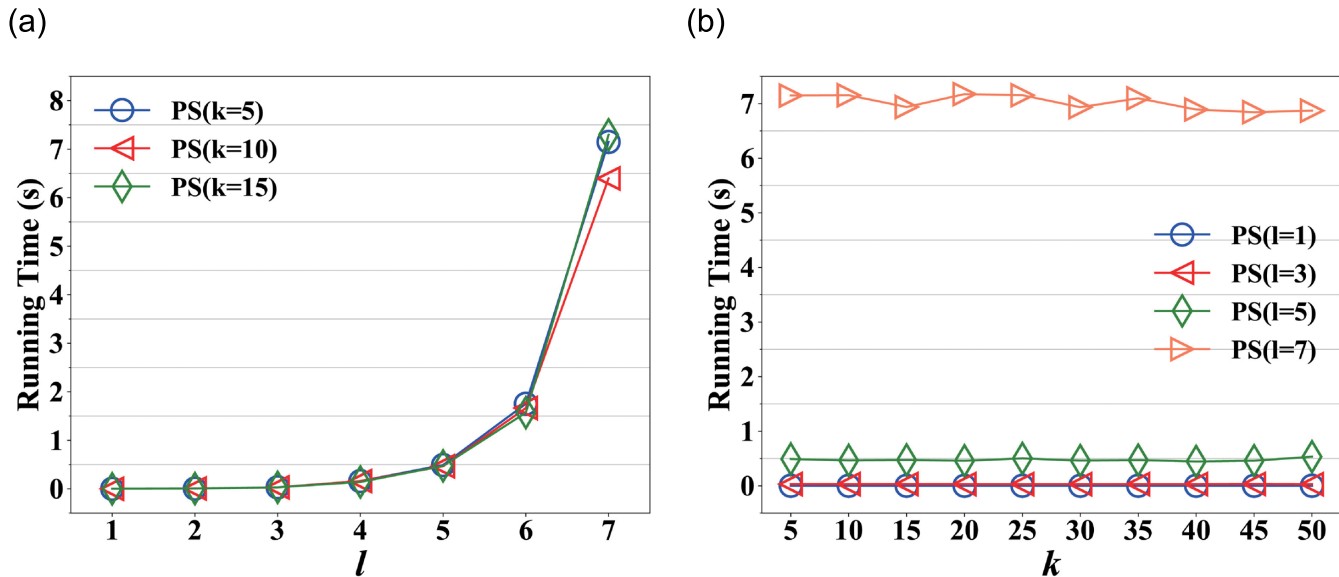

**Fig 4. Running time on varying $l$ and $k$ respectively in Nipic.** (a) Running time on varying $l$, (b) Running time on varying $k$.

(a)

(b)

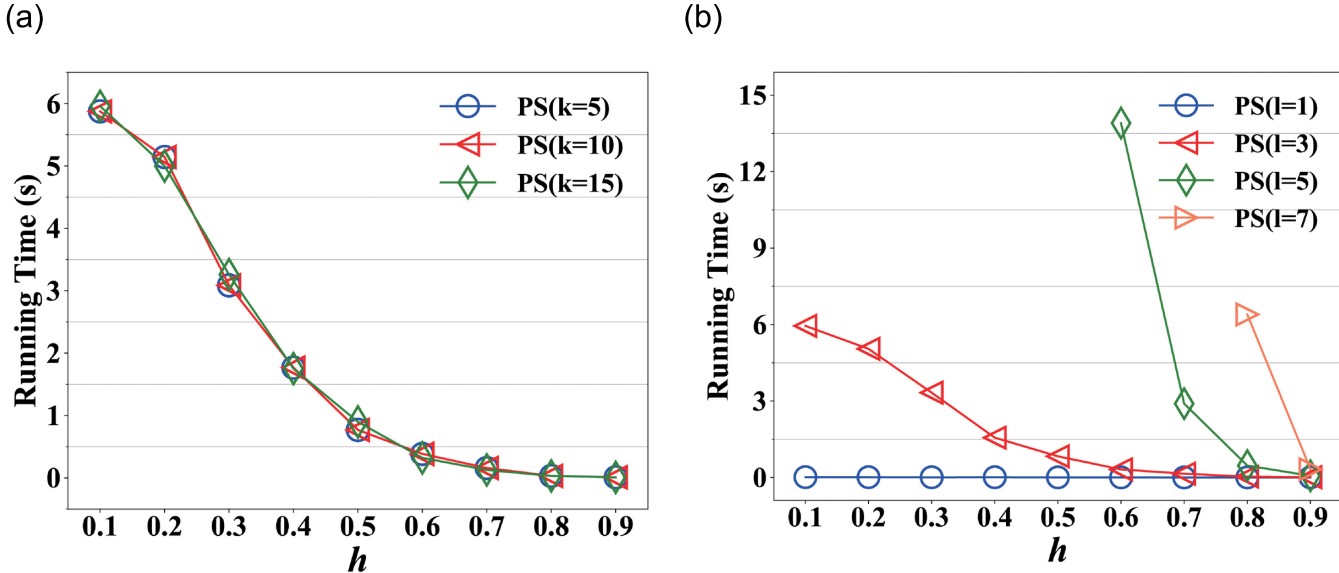

**Fig 5. Running time on varying $h$ in Nipic.** (a) Running time on varying $h$ at $l = 3$, (b) Running time on varying $h$ at $k = 10$.

Fig 4(b) shows the running time on varying $k$ in Nipic, where $l = 1, 3, 5, 7$. We observe that the running time almost remains stable as $k$ increases, which indicates time overhead does not change as $k$ increases. This is because the running time is affected by the sorting rather than the similarity calculation, and sorting overhead is almost negligible compared with the computational overhead. And the running time fluctuates significantly at $l = 7$, due to the instability of the machine.

Fig 5(a) shows the running time on varying $h$ in Nipic. We observe that the running time decreases as $h$ increases from 0.1 to 0.9. It drops rapidly from $h = 0$ to 0.6, and afterward, the slowly decreases as $h$ increases. Because of a larger $h$, the more noisy links will be removed in picture-tag network, which indicates the efficiency can be significantly improved after $h = 0.6$. So we set $h$ as 0.8 if not specified in other experiments.

Fig 5(b) shows the running time on varying $h$ in Nipic. The figure illustrates that the running time decreases as $h$ increases except for the curve of $l = 1$ remain stable, since we only consider direct in-links among nodes at $l = 1$ and the time change is minor as $h$ increases. At the same $l$, the larger network, the longer running time. The reason is that PictureSim iteratively calculates the similarity between pictures, which makes running time increase evidently.

## ImageNet

Fig 6(a) shows the MAP scores of PictureSim on varying $l$ in ImageNet. We observe that the MAP scores of PictureSim are relatively lower than that of Nipic. The reason is that each picture is described by only one tag, which fails to fully express the semantics information of the picture. Moreover, MAP scores irregularly fluctuate as $l$ increases, including the ranking of top 5, top 10 and top 15, because PictureSim searches all similar pictures at $l = 1$ and it has the same similarity scores. However, the returned ranking is different due to the sort algorithm. Fig 6(b) shows the MAP scores of PictureSim on varying $k$ in ImageNet. The results are similar to Fig 2(b), but the curve relatively fluctuates compared with Fig 2(b), and the difference between maximum and minimum is smaller than Nipic, the reason is as mentioned above.

(a)

(b)

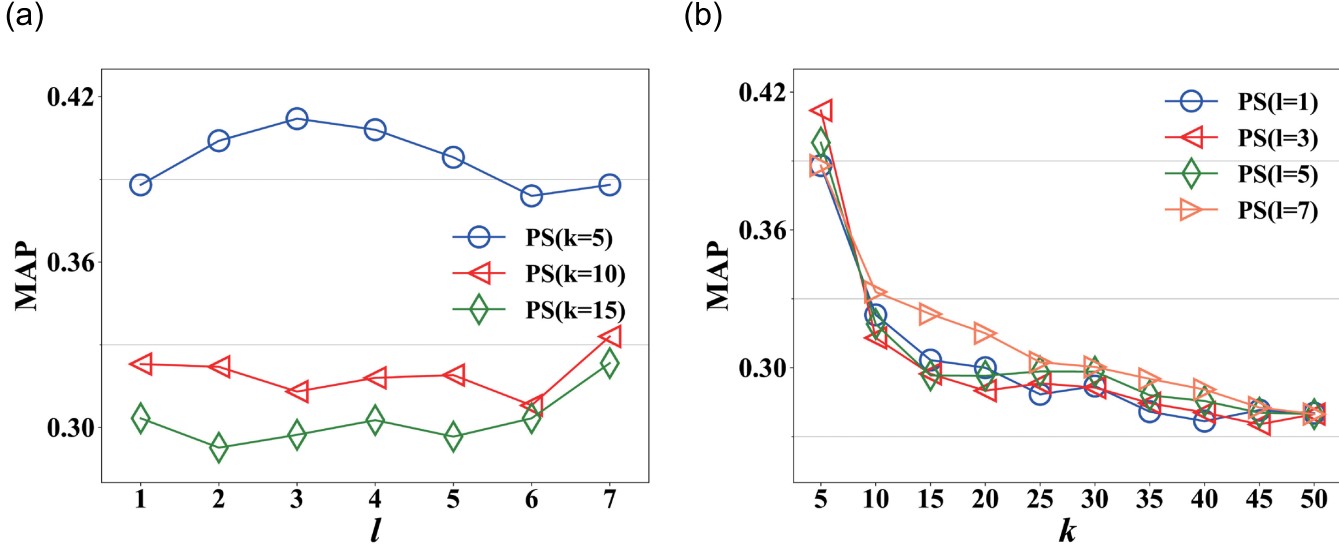

**Fig 6. MAP on varying *l* and *k* respectively in ImageNet.** (a) MAP on varying *l*, (b) MAP on varying *k*.

Fig 7(a) shows the running time on varying *l* in ImageNet, where *k* = 5, 10, 15. In which, the running time increases as *l* increases. But the time overhead is very small, especially ImageNet takes 0.004s at *l* = 7, while Nipic needs 7.3s. This is because each picture is described by only one tag, the picture-tag network is very sparse in ImageNet. Fig 7(b) shows the running time on varying *k* in ImageNet, where *l* = 1, 3, 5, 7. The result is similar to Fig 4(b), and the reason is as mentioned above. But the fluctuation of curve is relatively evident compared with Nipic. Because the time overhead of ImageNet is very small, so the time overhead of sort is relatively evident.

## Scalability

Fig 8 shows the scalability of PictureSim. We randomly select different *n* from Nipic and ImageNet, where *n* is the number of nodes. Fig 8(a) shows the running time slowly increases as *n*

(a)

(b)

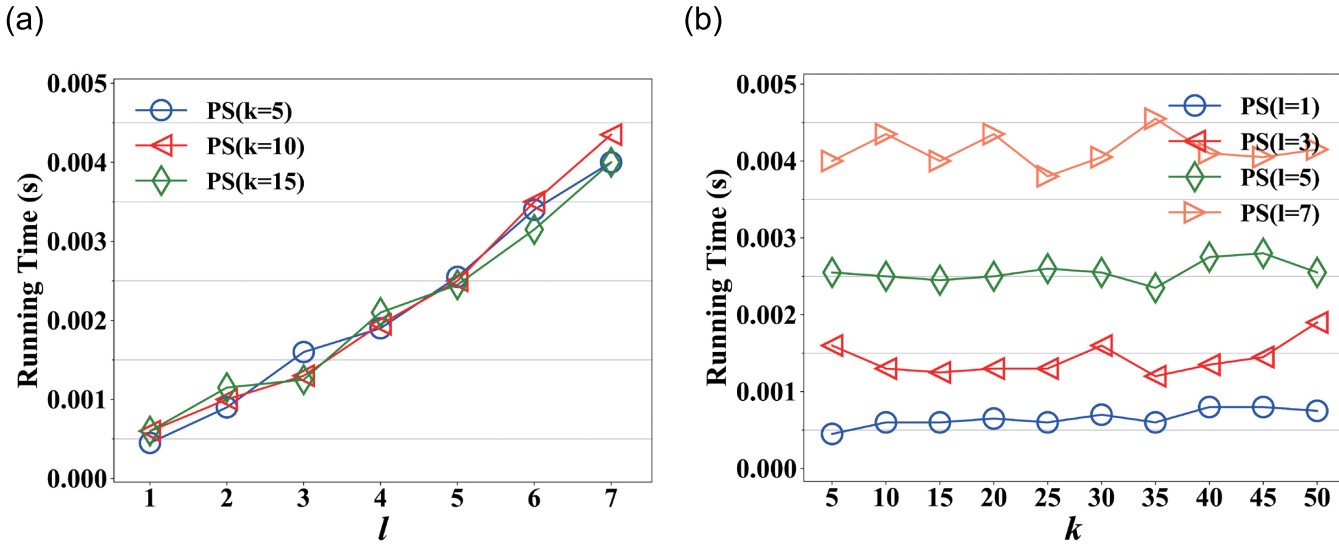

**Fig 7. Running time on varying *l* and *k* respectively in ImageNet.** (a) Running time on varying *l*, (b) Running time on varying *k*.

(a)

(b)

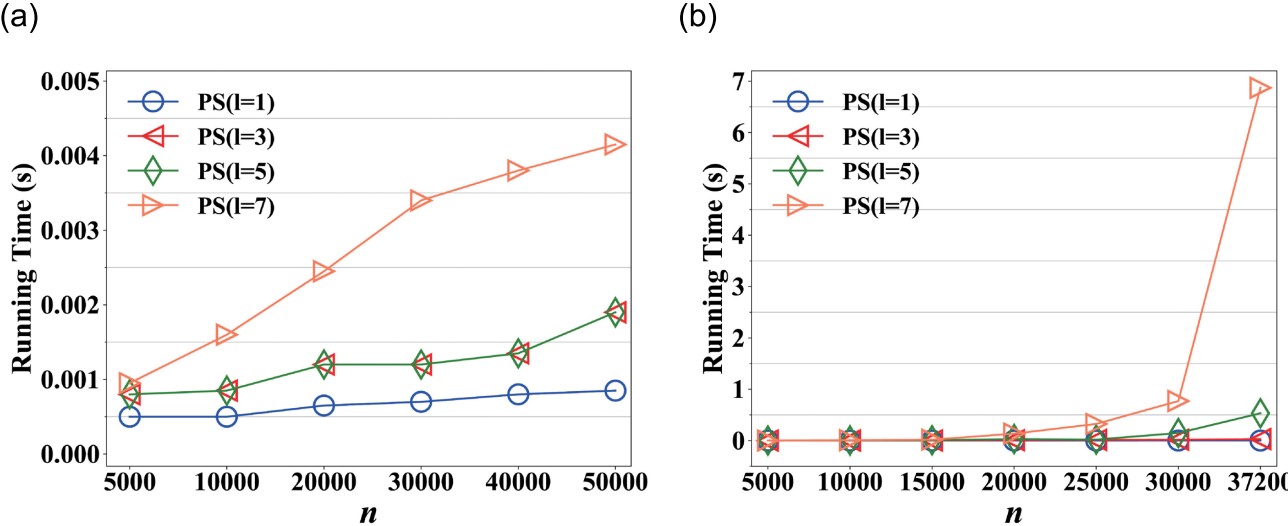

**Fig 8. Running time on varying n.** (a) ImageNet, (b) Nipic.

increases at $l$ = 1, 3, 5 and obviously increases at $l$ = 7. Because PictureSim iteratively computes similarity scores, as the network becomes larger, the running time obviously increases as $l$ increases. And due to the network is very sparse in ImageNet, the time overhead is very small.

Fig 8(b) shows the running time on varying $n$ in Nipic. The figure illustrates that the running time slowly increases as $n$ increases at $l$ = 1, 3, 5, and obviously increases at $l$ = 7, especially the $n$ varies from 30,000 to 37,200. The reason is that the picture-tag network becomes dense as $n$ increases, it traverses more paths as $l$ increases, which takes more time to obtain similar pictures. And the network is large, the time overhead increases exponentially as $l$ increases.

## Conclusion

This paper proposes a semantic similarity search method, namely PictureSim, for effectively searching similar pictures by building a picture-tag network. Compared with content-based methods, PictureSim can effectively and efficiently search similar pictures, which produces a better correlation with human judgments. Empirical studies on real datasets demonstrate the effectiveness and efficiency of our proposed approach. Future work will extend our approach to other datasets for effectively searching similar objects in other fields, because PictureSim is proposed for searching semantically similar pictures. Then, PictureSim requires $O(d^{2l})$ time cost, and the number of paths increases exponentially as path length increases, which makes computation expensive in terms of time and space and cannot support fast similarity search over large networks. So we will focus on reducing computational overhead to ensure timely response in large networks.

## Author Contributions

**Conceptualization:** Mingxi Zhang, Liuqian Yang.

**Data curation:** Liuqian Yang, Yipeng Dong, Jinhua Wang.

**Formal analysis:** Mingxi Zhang, Liuqian Yang.

**Funding acquisition:** Mingxi Zhang.

**Investigation:** Mingxi Zhang, Liuqian Yang.

**Methodology:** Mingxi Zhang, Liuqian Yang, Qinghan Zhang.

**Software:** Liuqian Yang.

**Writing – original draft:** Liuqian Yang.

**Writing – review & editing:** Liuqian Yang.

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
