## [Decision Letter · Decision Letter 0]

13 May 2021

PONE-D-21-06636

Picture Semantic Similarity Search Based on Bipartite Network of Picture-Tag Type

PLOS ONE

Dear Dr. Zhang,

Thank you for submitting your manuscript to PLOS ONE. After careful consideration, we feel that it has merit but does not fully meet PLOS ONE’s publication criteria as it currently stands. Therefore, we invite you to submit a revised version of the manuscript that addresses the points raised during the review process.

Based on the comments received from the reviewers and my own assessment, I suggest major revisions for the paper.

We look forward to receiving your revised manuscript.

Kind regards,

Thippa Reddy Gadekallu

Academic Editor

PLOS ONE

Journal Requirements:

2. We note that Tables 3 and 4 in your submission contain copyrighted images. All PLOS content is published under the Creative Commons Attribution License (CC BY 4.0), which means that the manuscript, images, and Supporting Information files will be freely available online, and any third party is permitted to access, download, copy, distribute, and use these materials in any way, even commercially, with proper attribution. For more information, see our copyright guidelines: http://journals.plos.org/plosone/s/licenses-and-copyright.

1.         You may seek permission from the original copyright holder of pictures in Tables 3 and 4 to publish the content specifically under the CC BY 4.0 license.

Reviewers' comments:

Reviewer's Responses to Questions

**Comments to the Author**

1. Is the manuscript technically sound, and do the data support the conclusions?

Reviewer #1: Yes

Reviewer #2: Yes

2. Has the statistical analysis been performed appropriately and rigorously? 

Reviewer #1: Yes

Reviewer #2: Yes

3. Have the authors made all data underlying the findings in their manuscript fully available?

Reviewer #1: Yes

Reviewer #2: Yes

4. Is the manuscript presented in an intelligible fashion and written in standard English?

Reviewer #1: Yes

Reviewer #2: Yes

5. Review Comments to the Author

Reviewer #1: 1. What are the limitations of the existing works?

2. The English language has to be polished.

3. Some of the recent works on ML/AI such as the following can be discussed in the paper: "Image-Based malware classification using ensemble of CNN architectures (IMCEC), A Novel PCA-Whale Optimization based Deep Neural Network model for Classification of Tomato Plant Diseases using GPU, Deep learning and medical image processing for coronavirus (COVID-19) pandemic: A survey, Hand gesture classification using a novel CNN-crow search algorithm".

4. Summarize the related works section in the form of a table.

5. Compare the current work with recent state-of-the-art.

6. Present a detailed analysis on the results obtained.

7. Present the computational complexity of the current work.

8. Discuss about the limitations of the current work in conclusion.

Reviewer #2: - Paper is well written. Author should add a little background of the study and limitations of the existing works and clearly explain the contributions at the end of the introduction.

- Qualities of figures are not good.

- Authors should add the most recent reference:

1 Cross corpus multi-lingual speech emotion recognition using ensemble learning, Complex & Intelligent Systems, 1-10

2) Byte-level object identification for forensic investigation of digital images, 2020 International Conference on Cyber Warfare and Security (ICCWS), 1-4

6. PLOS authors have the option to publish the peer review history of their article (what does this mean?). If published, this will include your full peer review and any attached files.

Reviewer #1: No

Reviewer #2: No

---

## [Author Response · Author response to Decision Letter 0]

8 Jul 2021

Response: 

After reading the journal requirements carefully, I tried our best to obtain permission from the original copyright holder of the pictures in table 3 and 4. However, the website has not copyright and should be asked permission by the authors. It is difficult to get permission from all authors, so we had to remove table 3 and 4. These changes will not influence the content and framework of the paper.

Response to reviewers: 

1. Response to Reviewer #1:

1) Comment: “What are the limitations of the existing works?”

Response: Thanks very much for your comments. We recognized that it is very important to discuss the limitations of existing works. For discussing the limitations, we wrote, “Practically, the measurement of picture similarity should be based on semantic information rather than visual features, which could cause a “semantic gap” between “semantic similarity” by human judgments and “visually similarity” by computer judgments. More precisely, picture semantic similarity is to answer the question “how similar are these two pictures?”. For example, if there are two pictures with different colors and backgrounds in “Cell phone” advertisements, which should be considered to be similar semantically, but they might be treated as dissimilar in visual features. On the other hand, different semantic pictures with similar visual features are judged to be similar pictures by the content-based methods. Due to the lack of semantic consideration, content-based metrics mainly focus on finding similar pictures in terms of visual features rather than semantics, which might neglect the expected similar pictures and deviate from the user’s intention.” (Line 30-41). 

2) Comment: “The English language has to be polished.”

Response: Thanks very much for your valuable comments regarding our paper. By your comments, we realize that the language should be polished, which is important to improve the quality of our manuscript. And we have invited a fluent English speaking colleague who has thoroughly improved the manuscript in terms of the English presentation. The langue was carefully polished, and it is much smoother now. Thank you very much again.

3) Comment: “Some of the recent works on ML/AI such as the following can be discussed in the paper: “Image-Based malware classification using ensemble of CNN architectures (IMCEC), A Novel PCA-Whale Optimization based Deep Neural Network model for Classification of Tomato Plant Diseases using GPU, Deep learning and medical image processing for coronavirus (COVID-19) pandemic: A survey, Hand gesture classification using a novel CNN-crow search algorithm”.”

Response: Thanks very much for your comments. By your suggestions, we added some discussions in the revised manuscript, “IMCEC [16] employs a deeper architecture of CNNs to provide different semantic representations of the picture, which makes it possible to extract features with higher qualities. [17] proposes a hybrid PCA–whale optimization-based deep learning model for the classification of picture, including transform picture dataset by one-hot encoding approach, reduce the dimensions of the transformed data by PCA and select the optimal features by WOA. [18] discusses the application of DL in medical image processing, which could realize the tracking, diagnosis and treatment of virus spread.” (Line 19-26).

Ref:

16. Danish Vasan SWBSQZ Mamoun Alazab. Image-Based malware classification using ensemble of CNN architectures (IMCEC). Comput Secur. 2020; 92: 101748.

17. Gadekallu T, Rajput D, Reddy P, Lakshman K, Bhattacharya S, Singh S, et al. A novel PCA–whale optimization-based deep neural network model for classification of tomato plant diseases using GPU. Journal of Real-Time Image Processing. 2020 06; p. 1–14.

18. Bhattacharya S, Reddy Maddikunta PK, Pham QV, Gadekallu TR, Krishnan S SR, Chowdhary CL, et al. Deep learning and medical image processing for coronavirus (COVID-19) pandemic: A survey. Sustainable Cities and Society. 2021; 65:102589.

4) Comment: “Summarize the related works section in the form of a table.”

Response: Thanks for your insightful comments and valuable suggestions. By your comments, we select the most related works to summarize in the form of a table, as show in Table 1, and we wrote, “Table 1 summarizes several picture similarity search methods, including content-based and link-based.” (Line 80-81).

5) Comment: “Compare the current work with recent state-of-the-art.”

Response: Thanks very much for your comments. After carefully studying your comment, we have compared the current work with recent state-of-the-art in the revised version, as we note, “Compared with the latest content-based metrics, link-based similarity measures could capture the semantic information of pictures based on a picture-tag network, while content-based methods mainly focus on searching similar pictures in visual features, which might neglect the expected similar pictures and deviate from the user’s intention. Moreover, the intuition of link-based methods is that “two pictures are similar if they are related to similar pictures”, which could search underlying similar pictures. For example, picture A is similar to picture B, and picture A is similar to picture C, so picture B is similar to picture C.” (Line 81-88).

6) Comment: “Present a detailed analysis on the results obtained.”

Response: Thanks very much for your comments. By your comments, we realize that it is important to analysis the results in detail. Correspondingly, we made more thorough discussions regarding the experimental results, as we wrote, “Table 2 shows the MAP scores of different metrics in Nicpic, and PictureSim sets l as 5. The MAP scores of PictureSim are obviously higher than that of traditional content-based methods with different k. For example at k = 15, PictureSim achieves average 0.599 MAE, while RD and HI yield average 0.119 MAE. This is because PictureSim computes similarity scores by the structure of context in the picture-tag network, while the traditional content-based approach considers the visual features, which often fails to reflect the semantic information in the user’s mind.” (Line 270-276), “Fig. 2(a) shows the MAP scores with varying l in Nipic, which clearly illustrates the effect of l in PictureSim. We observe that the MAP scores increase slowly as l increases from 1 to 5, because PictureSim not only considers direct in-links among nodes but also indirect in-links. After l = 5, the MAP scores become stable, and PictureSim converges to a stable state. So, the returned rankings would become stable empirically after the fifth iteration.” (Line 277-282), “Fig. 2(b) shows the MAP scores of PictureSim on varying k in Nipic. The MAP scores gradually decrease as k increases; it could achieve average 0.718 MAP at k = 5. This is because the higher similarity scores have a higher rank in the returned list. Generally, users are only interested in the top 10 similar pictures for a given picture, so PictureSim could achieve the user’s intention.” (Line 283-287), “Fig. 4(a) shows the running time on varying l in Nipic, in which, the running time increases slowly before l = 5 and increases rapidly after l = 5. This is because PictureSim also considers indirect in-links when searching similar pictures, it needs to traverse more paths as l increases. Fortunately, PictureSim could converge rapidly at l = 5 as shown in Fig. 2(a), which shows a good performance of the proposed approach.” (Line 307-312), “Fig. 4(b) shows the running time on varying k in Nipic, where l =1, 3, 5, 7. We observe that the running time almost remains stable as k increases, which indicates time overhead does not change as k increases. This is because running time is affected by the sorting rather than the similarity calculation, and sorting overhead is almost negligible compared with the computational overhead. And the running time fluctuates significantly at l = 7, due to the instability of the machine.” (Line 313-318), “Fig. 5(a) shows the running time on varying h in Nipic. We observe that the running time decreases as h increases from 0.1 to 0.9. It drops rapidly from h = 0 to 0.6, and afterward, the slowly decreases as h increases. Because a larger h, the more noisy links will be removed in picture-tag network, which indicates the efficiency can be significantly improved after h = 0.6. So, we set h as 0.8 if not specified in other experiments.” (Line 319-324), “Fig. 6(a) shows the MAP scores of PictureSim on varying l in ImageNet. We observe that the MAP scores of PictureSim is relatively lower than that of Nipic. The reason is that each picture is described by only one tag, which fails to fully express the semantics information of the picture. Moreover, MAP scores irregularly fluctuate as l increases, including the ranking of top 5, top 10 and top 15, because PictureSim searches all similar pictures at l = 1 and it has same similarity scores. However, the returned ranking is different due to the sort algorithm. Fig. 6(b) shows the MAP scores of PictureSim on varying k in ImageNet. The results are similar to Fig. 2(b), but the curve relatively fluctuates compared with Fig. 2(b), and the difference between maximum and minimum is smaller than Nipic, the reason is as mentioned above.” (Line 332-341), and “Fig. 7(a) shows the running time on varying l in ImageNet, where k =5, 10, 15. In which, the running time increases as l increases. But the time overhead is very small, especially ImageNet takes 0.004s at l = 7, while Nipic needs 7.3s. This is because each picture is described by only one tag, the picture-tag network is very sparse in ImageNet. Fig. 7(b) shows the running time on varying k in ImageNet, where l =1, 3, 5, 7. The result is similar to Fig. 4(b), and the reason is as mentioned above. But the fluctuate of curve is relatively evident compared with Nipic. Because the time overhead of ImageNet is very small, so the time overhead of sort is relatively evident.” (Line 342-349).

7) Comment: “Present the computational complexity of the current work.”

Response: Thanks very much for your comments. In our manuscript, we first analyzed the computational complexity of SimRank, as we wrote, “Though this can be saved time cost in on-line stage, expensive operations are required in the off-line stage, which involves O(n2) time cost and O(ld2n2) space cost at the l-th iteration, where n is the number of nodes in the network, d is the average degree of the nodes, and we set l from 1 to 7 in terms of time and cost overhead.” (Line 224-228). And then we analyzed the computational complexity of the PictureSim, as we wrote, “Therefore, PictureSim optimizes the efficiency of SimRank by TopSim algorithm without any preprocessing, which requires O(d2l) time cost in the on-line stage.” (Line 235-237).

8) Comment: “Discuss about the limitations of the current work in conclusion.”

Response: Thanks very much for your comments. After checking the paper carefully, we recognize that it is necessary to discuss the limitations of the current work, and we wrote, “Future work will extend our approach to other datasets for effectively searching similar objects in other fields, because PictureSim is proposed for searching semantically similar pictures. Then, PictureSim requires O(d2l) time cost, and the number of paths increases exponentially as path length increases, which makes computation expensive in terms of time and space and cannot support fast similarity search over large networks. So, we will focus on reducing computational overhead to ensure timely response in large networks.” (Line 369-375).

Response to Reviewer #2:

1) Comment: “Paper is well written. Author should add a little background of the study and limitations of the existing works and clearly explain the contributions at the end of the introduction.”

Response: Thanks for your insightful comments. By your comments, the background and limitations of the existing works are discussed, as we wrote, “Practically, the measurement of picture similarity should be based on semantic information rather than visual features, which could cause a “semantic gap” between “semantic similarity” by human judgments and “visually similarity” by computer judgments. More precisely, picture semantic similarity is to answer the question “how similar are these two pictures?”. For example, if there are two pictures with different colors and backgrounds in “Cell phone” advertisements, which should be considered to be similar semantically, but they might be treated as dissimilar in visual features. On the other hand, different semantic pictures with similar visual features are judged to be similar pictures by the content-based methods. Due to the lack of semantic consideration, content-based metrics mainly focus on finding similar pictures in terms of visual features rather than semantics, which might neglect the expected similar pictures and deviate from the user’s intention.” (Line 30-41). Then, we explain the contributions of our works more clearly, as we wrote in the introduction, “We build a picture-tag network by “description” relationships between pictures and tags. Initially, tags and pictures are treated as nodes, and relationships between pictures and tags are regarded as edges. Then, we propose a TF-IDF-based method to remove the noisy links by setting a threshold, which could measure whether a tag has good classification performance. We propose a link-based picture similarity search algorithm, namely PictureSim, for effectively searching similar pictures semantically, which considers the context structure to search underlying similar pictures in a network. And it could respond to the user’s requirement timely. We ran a comprehensive set of experiments on Nipic datasets and ImageNet datasets. Our results show that PictureSim achieves semantic similarity search between pictures, which produces a better correlation with human judgments compared with content-based methods.” (Line 99-111).

2) Comment: “Qualities of figures are not good.”

Response: Thanks very much for your valuable comments. Inspired by your suggestions, we improve the qualities of figures, such as the font size, icon size and so on. Figure 2 is example.

3) Comment: “Authors should add the most recent reference:

1) Cross corpus multi-lingual speech emotion recognition using ensemble learning, Complex & Intelligent Systems, 1-10

2) Byte-level object identification for forensic investigation of digital images, 2020 International Conference on Cyber Warfare and Security (ICCWS), 1-4”

Response: Thanks very much for your valuable comments, we have read the most recent reference carefully. Inspired by your suggestions, we added some references, as we wrote “[19] proposes an effective ensemble learning approach to identify and detect objects, which could achieve good accuracy on both with-in as well as cross-corpus datasets. [20] proposes a deep learning-based object detection approach, which utilizes ResNet to achieve fast robust and efficient object detection.” (Line 26-29).

Ref:

19. Zehra W, Javed AR, Jalil Z, Gadekallu T, Kahn H. Cross corpus multi-lingual speech emotion recognition using ensemble learning. Complex & Intelligent Systems. 2021 01; p. 1–10.

20. Javed AR, Jalil Z. Byte-Level Object Identification for Forensic Investigation of Digital Images; 2020. p. 1–4.

---

## [Decision Letter · Decision Letter 1]

10 Sep 2021

PONE-D-21-06636R1Picture Semantic Similarity Search Based on Bipartite Network of Picture-Tag TypePLOS ONE

Dear Dr. Zhang,

Thank you for submitting your manuscript to PLOS ONE. After careful consideration, we feel that it has merit but does not fully meet PLOS ONE’s publication criteria as it currently stands. Therefore, we invite you to submit a revised version of the manuscript that addresses the points raised during the review process.

We would like to inform you that the original editor was no longer available and has been replaced by another academic editor. Your manuscript has not been sent out for further review. The new editor conducted his own review (together with previous reviewers' comments) and concluded that your manuscript is suitable for publication after a minor revision. We observed that in the previous revision process you included some references suggested by both reviewers. We invite you to revise these modifications and include additional references only if you feel that doing so contributes to your manuscript. In addition to this point, we recommend that you consider making your Java code implementing PictureSim available for readers. As your main contribution is the proposition of a new computational method, we believe making an implementation available will significantly improve your work's impact.

We look forward to receiving your revised manuscript.

Kind regards,

Haroldo V. Ribeiro

Academic Editor

PLOS ONE

Journal Requirements:

Additional Editor Comments (if provided):

Minor misprints:

- "Content-based Image Retrieval*(CBIR)*" (there is a missing space)

- "they point to similar objects*.*."

- "Oriol et al. [38] *proposes*"

- "As *show* in Fig. 1,"

- "where c *ia* a constant"

- "Term frequency*(TF)*"

- "inverse document frequency*(IDF)*"

- "Minkowski Distance*(MD)* [42], Histogram Intersection*(HI)* [43] and Relative Deviation*(RD)*"

- "by Mean Average Precision*(MAP)*"

Reviewers' comments:

Reviewer's Responses to Questions

**Comments to the Author**

1. If the authors have adequately addressed your comments raised in a previous round of review and you feel that this manuscript is now acceptable for publication, you may indicate that here to bypass the “Comments to the Author” section, enter your conflict of interest statement in the “Confidential to Editor” section, and submit your "Accept" recommendation.

Reviewer #2: All comments have been addressed

2. Is the manuscript technically sound, and do the data support the conclusions?

Reviewer #2: Yes

3. Has the statistical analysis been performed appropriately and rigorously? 

Reviewer #2: Yes

4. Have the authors made all data underlying the findings in their manuscript fully available?

Reviewer #2: Yes

5. Is the manuscript presented in an intelligible fashion and written in standard English?

Reviewer #2: Yes

6. Review Comments to the Author

Reviewer #2: Accept!

7. PLOS authors have the option to publish the peer review history of their article (what does this mean?). If published, this will include your full peer review and any attached files.

Reviewer #2: No

---

## [Editor Report · Decision Letter 2]

12 Oct 2021

Picture Semantic Similarity Search Based on Bipartite Network of Picture-Tag Type

PONE-D-21-06636R2

Dear Dr. Zhang,

We’re pleased to inform you that your manuscript has been judged scientifically suitable for publication and will be formally accepted for publication once it meets all outstanding technical requirements.

Kind regards,

Haroldo V. Ribeiro

Academic Editor

PLOS ONE
---

## [Editor Report · Acceptance letter]

21 Oct 2021

PONE-D-21-06636R2 

Picture Semantic Similarity Search Based on Bipartite Network of Picture-Tag Type 

Dear Dr. Zhang:

I'm pleased to inform you that your manuscript has been deemed suitable for publication in PLOS ONE. Congratulations! Your manuscript is now with our production department. 

Kind regards, 

on behalf of

Dr. Haroldo V. Ribeiro 

Academic Editor

PLOS ONE